

# Maintenance of specificity in sympatric host-specific fig/wasp pollination mutualisms

Hua Xie[1,*], Pei Yang[2,*], Yan Xia[1], Finn Kjellberg[3], Clive T. Darwell[4] and Zong-Bo Li[1]

[1] Key Laboratory for Forest Resources Conservation and Utilization in the Southwest Mountains of China, Southwest Forestry University, Kunming, Yunnan, China
[2] Yunnan University of Chinese Medicine, Kunming, Yunnan, China
[3] CEFE, CNRS, Université de Montpellier, EPHE, IRD, Montpellier, France
[4] Biodiversity and Biocomplexity Unit, Okinawa Institute of Science and Technology Graduate University, Okinawa, Japan
[*] These authors contributed equally to this work.

## ABSTRACT

**Background**. Fig/wasp pollination mutualisms are extreme examples of species-specific plant-insect symbioses, but incomplete specificity occurs, with potentially important evolutionary consequences. Why pollinators enter alternative hosts, and the fates of pollinators and the figs they enter, are unknown.

**Methods**. We studied the pollinating fig wasp, *Ceratosolen emarginatus*, which concurrently interacts with its typical host *Ficus auriculata* and the locally sympatric alternative host *F. hainanensis*, recording frequencies of the wasp in figs of the alternative hosts. We measured ovipositor lengths of pollinators and style lengths in female and male figs in the two host species. Volatile organic compounds (VOCs) emitted by receptive figs of each species were identified using GC-MS. We tested the attraction of wasps to floral scents in choice experiments, and detected electrophysiologically active compounds by GC-EAD. We introduced *C. emarginatus* foundresses into figs of both species to reveal the consequences of entry into the alternative host.

**Results**. *C. emarginatus* entered a low proportion of figs of the alternative host, and produced offspring in a small proportion of them. Despite differences in the VOC profiles of the two fig species, they included shared semiochemicals. Although *C. emarginatus* females prefer receptive figs of *F. auriculata*, they are also attracted to those of *F. hainanensis*. *C. emarginatus* that entered male figs of *F. hainanensis* produced offspring, as their ovipositors were long enough to reach the bottom of the style; however, broods were larger and offspring smaller than in the typical host. Female figs of *F. hainanensis* failed to produce seeds when visited by *C. emarginatus*. These findings advance our current understanding of how these species-specific mutualisms usually remain stable and the conditions that allow their diversification.

Corresponding authors
Pei Yang, 521yangpei@163.com
Zong-Bo Li, lizb@outlook.com

## INTRODUCTION

Obligate species-specific pollination mutualisms are important and unique parts of natural ecosystems that facilitate efficient reproductive isolation between plant species (*Schiestl & Schlüter, 2009*). However, species-specificity is not absolute and numerous examples of alternative host-use by pollinators have been recorded (*Kawakita, 2010*; *Rasplus, 1996*; *Starr et al., 2013*; *Zhang et al., 2012*). The ongoing occurrence of such events without partner fidelity mechanisms that regulate species-specificity have the potential to undermine extant biodiversity patterns by either creating hybrid swarms among closely related species or by instigating hybrid-induced speciation events (*Coyne & Orr, 2004*). The proximate mechanisms that facilitate alternative host-use and those that help maintain species-specificity, alongside their potential evolutionary consequences, are largely unknown.

Partners in species-specific mutualisms have evolved sensory-mechanical filters including production of, and response to, particular mixes of volatile organic compounds (VOCs) that mediate host/pollinator encounter, and matching morphological traits that enforce specificity of the interaction (*Cornille et al., 2012*; *Okamoto, Kawakita & Kato, 2007*; *Raguso, 2008*). Such relationships have evolved between several plant and insect lineages, including *Ficus* (Moraceae) and their fig wasp pollinators (*Cook & Rasplus, 2003*), Yucca (Asparagaceae) and yucca moth pollinators (*Pellmyr et al., 1996*), and Phyllantheae and leafflower moth pollinators (*Kawakita, 2010*). In these systems, pollinators rear offspring exclusively within the reproductive structures of their host plants and they are the plants' sole pollinators (*Dufaÿ & Anstett, 2003*).

Strict specificity is predicted to lead to co-diversification over evolutionary timescales (*Silvieus, 2006*). However, numerous exceptions to this have been reported. They may involve multiple pollinators breeding on a single host, or, less frequently, a pollinator species locally interacting with different hosts (*Kawakita, 2010*; *Rasplus, 1996*; *Starr et al., 2013*; *Su et al., 2022*; *Zhang et al., 2012*). Among fig wasps, cases where a pollinator uses two hosts may result in interspecific hybridization among both hosts and wasps (*Molbo et al., 2003*; *Parrish et al., 2003*; *Wei et al., 2014*). While interspecific introgression may be a genetic dead-end if selection counters hybridization, it can also promote speciation (*Abbott et al., 2013*; *Su et al., 2022*; *Wang, Cannon & Chen, 2016*). Cases of incomplete specificity in species-specific mutualisms have received considerable attention (*de Vienne et al., 2013*; *Su et al., 2022*; *Whittall & Hodges, 2007*), but several important questions have not been addressed. For example, why would an exclusive pollinator associating with its own obligate host species interact with an alternative host plant that presumably provides sub-optimal breeding conditions? What are the consequences of this behavior for the reproductive success of pollinators? Most importantly, why does such behavior not result in the breakdown of species-specific mutualisms?

Among plants, barriers promoting reproductive isolation are typically classified as either pre- or post-pollination (*Baack et al., 2015*). In species-specific pollination systems, most studies have focused pre-pollination barriers with plants emitting distinctive volatile organic compound (VOC) mixes that attract specific pollinators (*Okamoto, Kawakita & Kato, 2007*; *Raguso, 2008*; *Scopece et al., 2013*; *Whitehead & Peakall, 2014*). However,

post-pollination barriers (including pre- and post-zygotic mechanisms) do occur in some cases and are typically mediated by pollen-stigma incompatibility, pollen competition, embryo abortion and hybrid sterility (*Scopece et al., 2013*).

*Ficus* species feature a unique globular inflorescence called a syconium or fig (*Janzen, 1979*). They are one of the largest genera of terrestrial plants and are keystone species in many tropical biomes (*Nason, Herre & Hamrick, 1998*). These figs usually undergo five developmental phases, including pre-female phase (A phase), female phase (B or receptive phase), interfloral phase (C phase), male-floral phase (D phase), and postfloral phase (E phase), over several weeks for a full developmental course. Only receptive figs allow pollinating wasps to enter the fig through the apical bracts forming the ostiole, which constitutes a mechanical filter to the pollinators. In the absence of pollinator visits, these receptive figs will cease development and abort. The pollinating wasps are mostly species-specific (*Cook & Rasplus, 2003*) with larvae developing within the fig's ovules (*Borges & Kjellberg, 2014*; *Jansen-González, Teixeira & Pereira, 2012*). In dioecious species, figs of male trees produce wasps and pollen, whereas figs of female trees produce only seeds. When figs become receptive, they emit floral scents comprising a mix of VOCs, which constitute the main signal used by a typically exclusive pollinator to locate its host plant. Floral scents exhibit quantitative and qualitative variation among host plant species in their VOC composition (*Souto-Vilarós et al., 2018*). Pollinating insects can therefore primarily rely on these specific signals as semiochemicals to locate their typical hosts, thus promoting species-specific interactions (*Cornille et al., 2012*; *Raguso, 2008*). A key determinant of oviposition success is relative ovipositor to style length. The egg is laid in the flower ovule between the inner integument and the nucellus. In male figs, female flowers have short styles, which allow easy access to the oviposition site by the ovipositor. On the contrary, on female figs these flowers have long styles and the oviposition site are, therefore, inaccessible (*Shi, Yang & Peng, 2006*).

Incomplete specificity of fig-wasp mutualisms has been documented in around 30–40% of cases (*Cook & Segar, 2010*; *Machado et al., 2005*; *Rasplus, 1996*; *Segar et al., 2014*; *Yang et al., 2015*). In most of these, multiple pollinators are associated with a widely distributed *Ficus* species in different parts of its range (*Bain et al., 2016*; *Cornille et al., 2012*; *Darwell, al Beidh & Cook, 2014*; *Rasplus, 1996*; *Rodriguez et al., 2017*; *Su et al., 2022*; *Yu et al., 2019*). However, in some cases, figs are entered not only by the typical pollinator but also by a pollinator from a sympatrically occurring *Ficus* species (*Machado et al., 2005*; *McLeish & Van Noort, 2012*; *Moe, Rossi & Weiblen, 2011*; *Ramírez, 1970*; *Souto-Vilarós et al., 2018*; *Yu et al., 2022*). Fig wasps may enter an alternative host because it produces floral scents similar to those of its typical host. However, little is known about how pollinating fig wasps perceive the odours of receptive figs and may be induced to visit them. Indeed, studies have typically focused on differences in VOC composition of floral odours among host plant species (*Ackerman, 1983*; *Starr et al., 2013*; *Sutton et al., 2017*; *Wang, Cannon & Chen, 2016*; *Zhang et al., 2012*) rather than on their similarities.

The evolutionary consequences of alternative pairings for figs and for wasps is also unknown. Limited recent gene exchange among *Ficus* species has been observed in cases where pollinators of one species regularly visit (normally at low frequencies) an alternative

host (*Machado et al., 2005*; *Su et al., 2022*; *Wang, Cannon & Chen, 2016*; *Wei et al., 2014*; *Yang et al., 2015*). This finding is consistent with data from studies of controlled pollinator introductions into alternative hosts. Introduction of four fig wasp species from other *Ficus* species into *F. turbinata* in Venezuela showed that while these wasps produced offspring, no viable seeds were produced (*Ramírez, 1970*). In dioecious *F. montana*, its pollinator, *Kradibia tentacularis*, produced no progeny when introduced into male figs of *F. asperifolia*, but female figs of *F. asperifolia* produced viable seeds after introduction of *K. tentacularis* bearing *F. montana* pollen (*Ghana, Suleman & Compton, 2015*). Few experimental studies of pollinating wasps entering alternative hosts have examined the fates of resulting pollinator offspring and fig seeds. Where wasp offspring are produced in alternative hosts, no information exists on resultant morphological traits and whether fitnesses are affected. It is also unknown whether pollinators and plants might also hybridize.

The fig pollinating wasp association between *F. auriculata* and *Ceratosolen emarginatus*, provides an ideal study system in which the pollinators occasionally visit figs of the alternative host, *F. hainannensis* (Fig. 1; (*Yang et al., 2012*); further see study system below). Here we address the following questions: (i) What are the frequencies of pollinator visitation on alternative hosts in natural populations? (ii) What are the differences and the similarities in profiles of the VOCs produced by receptive figs of the two host species? (iii) Do receptive figs of both hosts attract *C. emarginatus*? (iv) Do the two fig species share semiochemicals that are electrophysiologically active in *C. emarginatus*? (v) Does relative ovipositor length of *C. emarginatus* among both typical and alternative hosts vary? (vi) What are the potential evolutionary consequences of entry by *C. emarginatus* into an alternative host, for both wasps and figs?

## MATERIALS & METHODS

### *Ficus* choice in study sites

This study was conducted in Xishuangbanna, Yunnan, southwestern China. *Ficus auriculata* is distributed in moist valleys in rain forests, whereas *F. hainanensis* is distributed in limestone areas along rivers. We chose the natural habitats of the two species as study sites, the former in the rain forest in XTBG (Xishuangbanna Tropical Botanical Garden) in Menglun town, and the latter along the Mengxing River in Mengxing town. All study trees are non-cultivated and their identity was previously validated by genetic data presented in *Wei et al. (2014)*. We use the same nomenclature as in that study. The two study sites are 10 km apart.

### General biology and pollinator resource of the fig-wasp pairing study system

The two study *Ficus* species, *F. auriculata* and *F. hainanensis*, are dioecious species (subsection *Neomorphe*, section *Sycomorus*, subgenus *Sycomorus*). *F. auriculata* exhibits a heart-shaped to round leaves (15–55 × 10–27 cm), and its figs are produced on cauliforous branches on the main trunk and main branches. They are pear-shaped, 3–6 cm in diameter, and green-reddish when unmature (A-D phase). *F. hainanensis* is distinguishes from *F. auriculata* by its narrower obovate to elliptic leaves (12–25 × 6–23 cm). The figs are borne

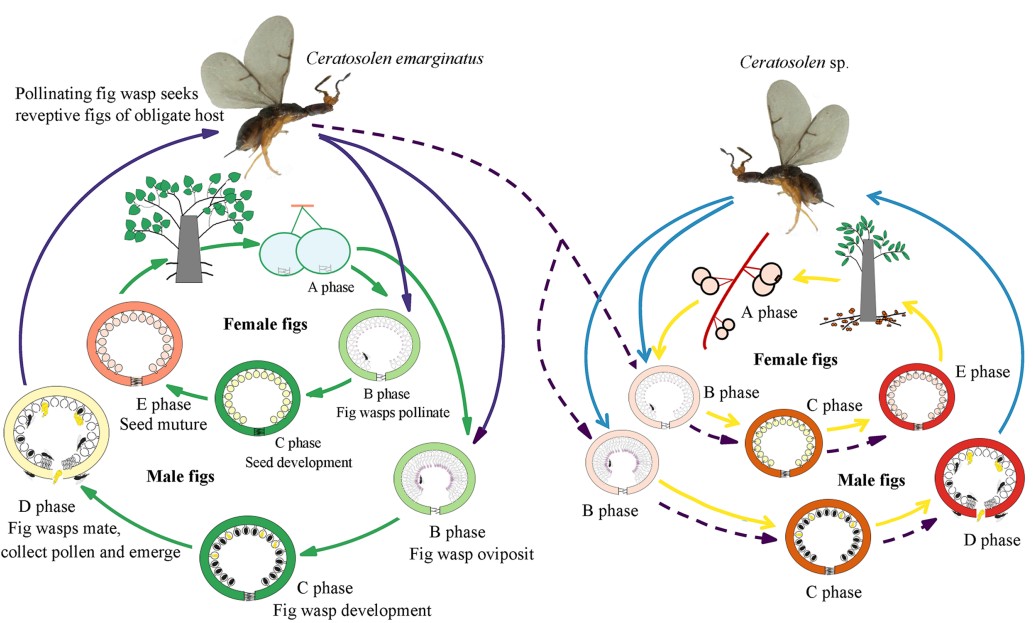

**Figure 1** Illustration of sharing the pollinator between two distinct host *Ficus* species.

on long stoloniflorous branches expanding from the base of the trunk. The pear-shaped, 1–2 cm diameter figs are sparsely speckled with globose tuberculate over the reddish surface (A-E phase). They usually bear figs in December every year at our study site. Their fruiting period is synchronous within species. *F. auriculata* is typically pollinated by the species traditionally identified as *Ceratosolen emarginatus* Mayr (Clade 1 in *Wang, Cannon & Chen, 2016*), and occasionally by what could be another *Ceratosolen* species involved in two study species and *F. oligodon* (Clade 3 in *Wang, Cannon & Chen, 2016*), and produced offspring in *F. hainanensis* (Fig. 1; *Yang et al., 2012*). However, *F. hainanensis* is almost exclusively pollinated by a closely related species, *Ceratosolen* sp. (Clade 2 in *Wang, Cannon & Chen, 2016*), which also enter figs of *F. auriculata* successfully, but leads to a high fig abortion rate (88.3%, *Yang et al., 2012*). This is the reason we selected *F. auriculata - C. emarginatus* as a study system. To distinguish *C. emarginatus* from *Ceratosolen* sp., we examined their external morphology (Fig. 2): among other traits, most obviously, the appendages of the mandibulae are notably closer to the maxilla in *C. emarginatus* than in *Ceratosolen* sp. To further control which wasp species were used in our experiments (*e.g.*, bioassay, GC-EAD, and introduction), we introduced a single female wasp of *C. emarginatus* into each fig so that we could use the resulting offspring for experimentation without need for species identification. These offspring were temporary stored into a 120-mesh nylon bag (to prevent escape) at room temperature after emergence from the male-phase figs.

## Investigation of pollinator host use in natural populations

We investigated the numbers of *C. emarginatus* and *Ceratosolen* sp. that had entered and produced offspring into the figs of *F. auriculata* and *F. hainanensis* by collecting post receptive figs, five days after the end of receptivity and male-floral phase, respectively. For

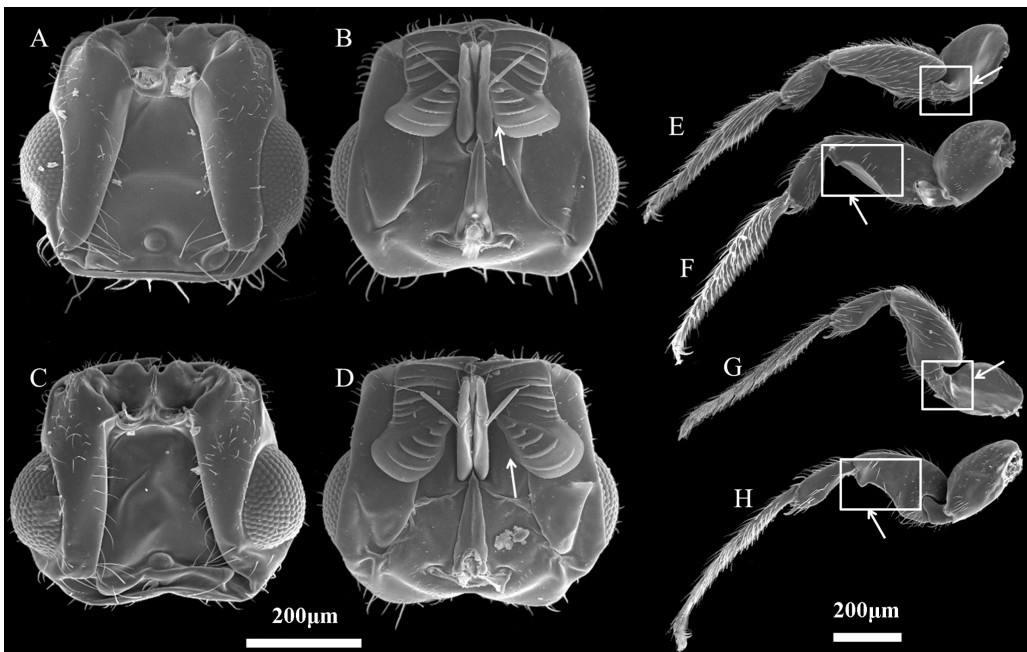

**Figure 2** **Characteristics of *Ceratosolen emarginatus* and *Ceratosolen* sp.** (A) Dorsal side of head of *C. emarginatus*; (B) ventral side of head of *C. emarginatus*; (C) dorsal side of head of *Ceratosolen* sp.; (D) ventral side of head of *Ceratosolen* sp.; (E) dorsal side of hind leg of *C. emarginatus*; (F) ventral side of hind leg of *C. emarginatus*; (G) dorsal side of hind leg of *Ceratosolen* sp.; (H) ventral side of hind leg of *Ceratosolen* sp. Scale: 200 μm. Note that, when the mandibulae are closed, the appendages of the female mandibulae are close to the maxilla in *C. emarginatus* while the mandibular appendages are more divergent from each other and hence more separated from the maxilla in *Ceratosolen* sp. Furthermore, the hind legs of *C. emarginatus* have a large ventral tooth in the coxa and a sharp tooth in the femur, thus differing from those of *Ceratosolen* sp.

the former, we collected at least 20 post-receptive figs from each of four male and four female trees. For the latter, we collected 20 male-phase figs per male tree at the same tree about two months later. Each post-receptive fig was cut open and effective foundresses were collected from the cavity and preserved in 75% alcohol for morphological species identification and counting. Every male-phase fig was put into a nylon bag (120 mesh) shortly before wasp emergence. Once wasps had emerged into the bag, ten wasps per bag were randomly selected and preserved in 75% alcohol for subsequent species identification. Combing with the aforementioned morphological criteria (Fig. 2), we finally identified these wasp species using a key table to the species of fig wasps associated with the *Ficus* subgenus *Sycomorus* in Xishuangbanna, then counted and classified them into each host species.

## Behavioral bioassays

To test whether *C. emarginatus* was preferentially attracted to its typical over alternative figs, behavioral choice experiments were performed using a Y-tube olfactometer (ID:1.5 cm, length of each arm nine cm, stem eight cm; for further information see *Chen et al., 2009*). Two arms of the Y-tube were each connected to a polyethylene terephthalate bag (Toppits®
GmbH, Germany) containing a source of odour. Airflow was pumped by a mini-vacuum pump (Xinweicheng®; Xinweicheng Machinery & Electric Co., Ltd, Chengdu, China) through the bags into the arms of the Y-tube, after being purified by passing through an activated charcoal filter. The flow rate through each arm was maintained at 100 ml/min. A wasp was deposited in the third arm of the olfactometer for behavioral observation. Preliminary experiments showed that *C. emarginatus* was equally attracted by male and female figs of *F. auriculata* (male *Vs* female = 18 *Vs* 16; $\chi^2 = 0.118$, $P = 0.732$), so we used only male figs as a source of odours.

To prepare the odour sources, pre-receptive stage figs of *F. auriculata* and *F. hainanensis* were enclosed in nylon bags to protect them from oviposition by wasps until the figs became receptive. Male figs were collected just before pollinator emergence to obtain freshly emerged fig wasps, and only one wasp was selected from each fig. To test the response of *C. emarginatus* to receptive figs of typical and alternative host, three treatments were carried out: receptive figs of *F. auriculata versus* air, receptive figs of *F. hainanensis versus* air, and receptive figs of *F. auriculata versus* receptive figs of *F. hainanensis*. Because of differences in fig size between species, equal weights of figs of the two hosts were used (*F. auriculata*: 5–6 figs, diameter range: 36.55–59.41 mm; *F. hainanensis*: 22–28 figs, diameter range: 12.35–21.70 mm). All bioassays were performed in a darkened room, between 10:00–12:00 h, within three hours after collection of fresh figs from trees. Each of the female fig wasps tested was positioned at the entrance to the stem of the olfactometer, and the arm it selected was recorded as well as the time to decision. If the wasp did not make a decision within 5 min, it was excluded from the total number counted and from statistical analysis. After testing five successive wasps, Y-tube arms were reversed to cancel out any orientation-bias effect between the two arms. Each wasp individual was tested only once, and, after ten successive wasps, the Y-tube was replaced with a new one to avoid any influence of residual materials remaining in the apparatus. Three treatments were repeated with total 31, 38 and 41 fig wasps that made a decision.

## Comparison of VOCs emitted by two host species

VOCs were collected using the dynamic headspace technique (*Chen et al., 2009*). Small pre-receptive figs for each sex of *F. auriculata* and *F. hainanensis* were enclosed in nylon bags on trees to prevent wasps from entering and ovipositing. When the figs had reached receptive phase (*F. auriculata*: 4-8 figs, *F. hainanensis*: 20-35 figs), they were enclosed in polyethylene terephthalate bags for the collection of VOCs. Airflow, purified by passing through a filter of activated charcoal (20-40 mesh, Supelco®; Sigma-Aldrich, St, Louis, MO, USA), was maintained through the bag by a mini-vacuum pump connected to the entrance by flow-meters with a flow rate of 300 ml/min, while a VOC trap containing 80 mg Porapak® Q adsorbent (80-100 mesh, Supelco®; Sigma-Aldrich, St, Louis, MO, USA) was connected to the exit of the bag at a flow rate of 300 ml/min. To check for possible contaminant compounds sampled during collection, empty bags were used as blanks for extraction by means of the same dynamic headspace technique and equipment. VOC collection was performed for four hours from 10:00 to 14:00, the period of the day when fig wasps are most active. Three repeats were performed for each sex of each tree

species. After VOC collection, the adsorbents were eluted three times with a total of 500 μl of dichloromethane and concentrated down to 100 μl with 99.99% $N_2$. Then, two internal standards (octane and dodecane, at 200 ng/ μl) were added to every sample prior to gas chromatography.

All VOC samples were analysed using gas chromatography-mass spectrometry (GC-MS, 7890A-5975C; Agilent Technologies, Santa Clara, CA, USA) with an HP-5MS column (30 m, ID: 250 μm, film thickness 0.25 μm). Helium was used as carrier gas at a flow rate of 1ml/min. The injector split vent was set at a ratio of 1:4 and the injector temperature was 250 °C. Oven temperature was set at 40 °C, and then programmed to rise to 150 °C at a rate of 3 °C/min, then at 10 °C/min to 260 °C, and finally temperature was maintained at 260 °C for 5 min. Compound identification was based on comparison of retention times (RT), matching of the mass spectra with the NIST 08 MS library, and Kovats retention indices (RI) taken from both the NIST Chemistry Web Book (http://webbook.nist.gov) and the RI database (*Adams, 2007*). Where available, we used synthetic compounds as a more precise reference (see Table S1).

### Electrophysiological responses of *C. emarginatus* to VOCs of the two host species

To identify which VOCs of the receptive fig odours were detected by the wasps and thus constituted candidate semiochemicals, we performed electrophysiological tests. The responses of *C. emarginatus* antennae to odours from receptive figs of *F. auriculata* and *F. hainanensis* were recorded using gas chromatography-electroantennography (GC-EAD; Agilent, USA, Syntech, Netherlands). VOCs were collected for injection in the GC-EAD as for the VOC analysis procedures except that we extended the collection duration to six hours in order to extract larger quantities of VOCs. The GC program was the same as that used for the analysis of VOCs presented above. A head with an antenna was placed on a micro-operating platform (MP-15; Syntech, Netherlands), and two glass electrodes filled with saline solution (NaCl, 4 g; $Na_2HPO_4$ 0.57 g; $KH_2PO_4$, 0.1 g; KCl, 0.1 g in 500 ml distilled water; pH 7.4) were connected to the distal tip of the antenna and to the antennal scape. Antenna depolarization was recorded using the Electroantennography version 2.5 software package (Syntech, Netherlands). Three antennae from *C. emarginatus* were tested for *F. auriculata* VOCs and three for *F. hainanensis* VOCs.

### Relative ovipositor to style length

As pollinating wasps oviposit by inserting their ovipositors into the female styles of pistillate flowers, we investigated matching relationship between style length in both host *Ficus* species and ovipositor length of the pollinator *C. emarginatus*. We enclosed the pre-receptive figs of *F. auriculata* and *F. hainanensis* in nylon bags (120 mesh) to prevent wasp entrance. When the figs became receptive to the pollinators, we collected 32 figs from four *F. auriculata* male trees and 30 figs from three female trees; we collected 30 figs from three *F. hainanensis* male trees and 31 figs from three female trees. Then 20 pistillate flowers per fig were randomly selected to measure style length. Ninety male figs of *F. auriculata* were collected and placed in separate bags before female wasp emergence.

After emergence we selected one *C. emarginatus* from every fig and measured the length of the ovipositor, which was excised from the wasp abdomen and removed the ovipositor sheath. All measurements of style length of figs and ovipositor length of pollinators were conducted using a micrometer in a dissection microscope (Olympus SZX12-3141, Tokyo, Japan).

### *C. emarginatus* introduction experiments

Pre-receptive figs of *F. auriculata* and *F. hainanensis* were enclosed in large nylon bags (120 mesh) to prevent wasp oviposition. When these figs reached receptive phase, one *C. emarginatus* emerging from *F. auriculata* was introduced into each fig. Wasps were introduced into at least 20 figs for each tree. We chose three trees of each sex for each host species. After introduction, figs were re-enclosed in large nylon bags until just before wasps emerged from the figs. The figs were then removed from the tree and enclosed in individual nylon bags. All emerging wasps were preserved in 75% alcohol for subsequent counting and measurements. According to morphological convergence produced by the selective pressures from the ostiole (*Liu, Yang & Peng, 2011*; *van Noort & Compton, 1996*), three traits representing fig wasp size, including head width, thorax width, and ovipositor length, were measured on the offspring for comparison with the foundresses. At least five wasps from each fig for each tree were measured. All measurements were carried out under a stereomicroscope.

### Data analysis

Data analyses were mostly performed in R version 4.0.5 (*R Core Team, 2020*). For VOC analyses, non-metric multi-dimensional scaling (NMDS) methods were conducted using the Vegan package and the Bray-Curtis distance was used to find the best two-dimensional representation of the distance matrix. A Permutational Multivariate Analysis of Variance (PERMANOVA) was used to compare the VOC composition between *F. auriculata* and *F. hainanensis*. Chi-square tests were used to determine whether pollinators showed preferences for their typical or alternative host. Mann–Whitney U tests were used to test the time that pollinators took to make a choice in the behavior-choice experiment. ANOVAs or Kruskal-Wallis test with unequal variances were used to examine differences in style length among figs and trees as well as head width, thorax width and ovipositor length between foundresses and offspring in the two treatments. Style length, ovipositor length and number of offspring in the two treatments were also compared using Mann–Whitney U tests.

### Field study permissions

The following information was supplied relating to field study approvals (*i.e.,* approving body and any reference numbers): the agonid wasps, *C. emarginatus* and *Ceratosolen* sp., are common insects and collection permitted by the leader of Department of Horticulture and Landscape, Xishuangbanna Tropical Botanical Garden, Chinese Academy of Science.

**Table 1** Number of foundresses entering receptive figs of two host *Ficus* species and offspring production at the male-floral phase.

| *Ficus* species | Number of foundresses entering receptive figs | | | | | Offspring produced by figs at the male-floral phase | | |
|---|---|---|---|---|---|---|---|---|
| | Sample | Occurrence rate | Range | Mean | Total | Sample | Occurrence rate | Total |
| F. auriculata | | | | | | | | |
| Male figs | 86 | 100% | 1–38 | 11.4 | 977 | 80 | 100% | 793 |
| Female figs | 88 | 100% | 1–25 | 11.6 | 1019 | – | – | – |
| F. hainanensis | | | | | | | | |
| Male figs | 59 | 40.7% | 1–14 | 1.5 | 90 | 60 | 18.3% | 30 |
| Female figs | 60 | 18.3% | 1 | 0.2 | 11 | – | – | – |

## RESULTS

### Frequency of *C. emarginatus* entering the alternative host and offspring production

We collected a total of 2,079 wasps (1,036 wasps in 86 male figs and 1,043 in 88 female figs) that had entered 174 receptive figs of *F. auriculata* (Table 1). Among these, each fig contained at least one *C. emarginatus* (one exception), but the total number of foundresses entering into a fig showed a significant variation (male fig:1-38 wasps, female fig: 1–25 wasps). There was no difference in average of foundresses numbers between male (11.4 ± 7.8) and female fig (11.6 ± 6.5). Totally, the proportions of *C. emarginatus* entering figs of *F. auriculata* were 94.31% in male figs and 97.69% in female figs, respectively. We collected a total of 931 wasps (757 in 60 male figs and 174 in 60 female figs) that had entered 120 receptive *F. hainanensis* figs. There was a small number of *C. emarginatus* (11.89% in male figs and 6.32% in female figs). Among these, 24 out of 60 male figs and 11 out of 60 female figs contained *C. emarginatus*. The total number of *C. emarginatus* in a male fig was larger than in a female fig (male fig: 1–14 wasps, female fig: 1 wasp). Results for offspring production were qualitatively similar to those for fig visitation, but the differences between typical and alternative hosts were more extreme (Table 1). In *F. auriculata*, 793 *C. emarginatus* offspring from 80 male figs (occurring rate of 100%) were found while from 11 figs from 60 male *F. hainanensis* figs (occurring rate of 18.3%), there were of total 30 *C. emarginatus* individuals.

### Pollinator bioassays

Female *C. emarginatus* individuals were strongly attracted by receptive figs of their typical host, *F. auriculata*, when confronted with a choice between it and air or the alternative host *F. hainanensis*. The wasps preferred *F. hainanensis* receptive figs over air (Fig. 3). When the wasps were given a choice between *F. auriculata* and *F. hainanensis*, there was a reduction in the proportion of wasps choosing *F auriculata* figs when compared to the *F. auriculata* versus air choice ($\chi^2 = 8.656$, $P = 0.003$). When given a choice between *F. auriculata* odour and air, *C. emarginatus* spent the shortest mean recorded time before entering a branch of the olfactometer (39.91 ± 22.35 s, $P = 0.036$). *C. emarginatus* preferred *F. auriculata* odour when given a choice between it and *F. hainanensis*, with the time taken independent of final arm choice (*F. auriculata*: 54.78 ± 36.41 s, *F. hainanensis*: 56.75 ± 46.60 s, Z = −0.078,

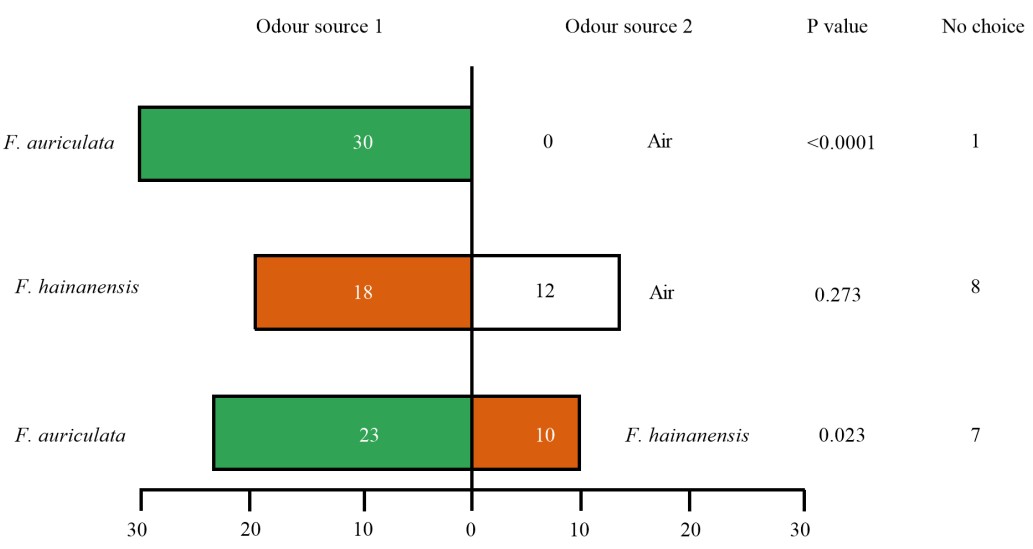

| | Odour source 1 | | Odour source 2 | P value | No choice |
|---|---|---|---|---|---|

*F. auriculata* — 30 | 0 | Air | <0.0001 | 1

*F. hainanensis* — 18 | 12 | Air | 0.273 | 8

*F. auriculata* — 23 | 10 | *F. hainanensis* | 0.023 | 7

**Figure 3** Bioassays of female *Ceratosolen emarginatus* responses to receptive figs of *Ficus auriculata* and *F. hainanensis* performed using Y-tube olfactometer tests.

$P = 0.938$). In the *F. hainanensis versus* air treatment, *C. emarginatus* took longer before entering an arm ($63.99 \pm 38.29$ s) in comparison to the *F. auriculata versus* air treatment ($Z = -2.641$, $P = 0.008$).

## Comparison of the VOCs emitted by the two *Ficus* species

We identified a total of 78 VOCs in scents emitted by receptive figs of *F. auriculata* and *F. hainanensis*. VOCs emitted by receptive figs did not differ significantly between sexes within species (Fig. 4, PERMANOVA; for *F. auriculata*, $F = 2.68$, $P = 0.062$, for *F. hainanensis*, $F = 3.95$, $P = 0.100$; Table S1). Thirty-four VOCs were shared between the odours produced by the two fig species, and among these, ten VOCs were abundant (>5%) in one or both species. In particular, the relative ratios of $\beta$-funebrene were high in both species (*F. auriculata*: $12.22 \pm 5.50\%$, *F. hainanensis*: $26.83 \pm 5.48\%$). Nevertheless, the complement of VOCs produced by *F. auriculata* and *F. hainanensis* were distinguishable (PERMANOVA, $F = 11.297$, $P = 0.002$).

## Electrophysiological responses of *C. emarginatus* to VOCs of the two *Ficus* species

*C. emarginatus* presented electroantennographic responses to nine compounds in the odours from *F. auriculata* and to six in the odours from *F. hainanensis* (Fig. 5). Among these compounds, $\alpha$-copaene and $\beta$-funebrene were produced by both host trees. These two compounds represented 38% and 31% of the scents emitted by male and female *F. hainanensis* figs respectively.

## Matching of style length and ovipositor length

Style lengths were bimodally distributed in both *F. auriculata* and *F. hainanensis*, while they significantly differed among figs and trees (Table 2; Fig. 6). Style lengths of figs

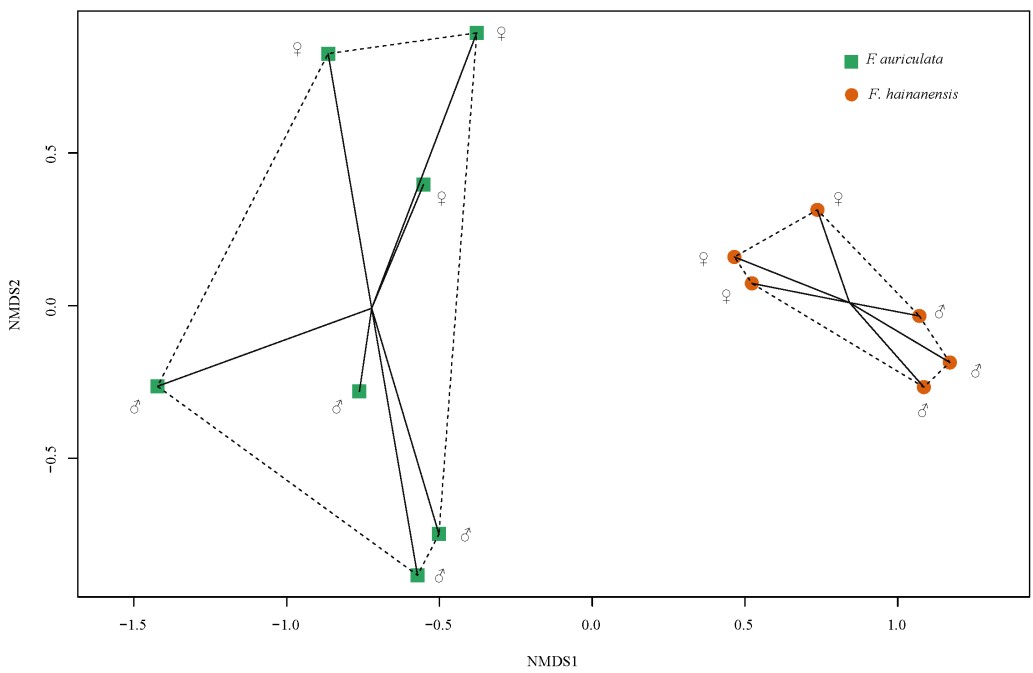

**Figure 4** **Non-metric multi-dimensional scaling of the relative proportions of VOCs emitted by receptive figs of *Ficus auriculata* and *F. hainanensis* based on Bray–Curtis dissimilarity index (stress = 0.062).** The tendency for a slight difference in receptive fig odour between sexes is non-significant whereas the difference between species is significant.

of female trees (*F. auriculata*: 2.03 ± 0.75 mm, *F. hainanensis*: 1.54 ± 0.18 mm) were much longer than in figs of male trees in both *Ficus* species (*F. auriculata*: $Z = -25.534$, $P < 0.001$; *F. hainanensis*: $Z = -30.292$, $P < 0.001$). Style length in figs of male trees of *F. auriculata* (0.97 ± 0.11 mm) was around 0.05 mm longer than that for figs of male trees of *F. hainanensis* (0.92 ± 0.05 mm) ($Z = 9.295$, $P < 0.001$). Ovipositor of *C. emarginatus* was 1.19 ± 0.09 mm ($n = 90$) in length, which were very similar to the styles of male figs from the two *Ficus* hosts (Fig. 6), but shorter than the styles in female figs (*F. auriculata*: $Z = 11.10$, $P < 0.001$; *F. hainanensis*: $Z = -8.93$, $P < 0.001$).

## Consequences of introduction of *C. emarginatus*

Single *C. emarginatus* females produced significantly fewer offspring in *F. auriculata* figs than in *F. hainanensis* figs ($Z = -4.490$, $P < 0.001$; Table 3). When *C. emarginatus* was introduced into female figs of *F. hainanensis*, all treated figs aborted. There was no significant fig and tree effect on wasp-size of female *C. emarginatus* reared in *F. auriculata* figs (head width: $P = 0.097$ and $P = 0.076$, thorax width: $P = 0.709$ and $P = 0.512$; ovipositor length: $P = 0.073$ and $P = 0.510$) and *F. hainanensis* (head width: $P = 0.204$ and $P = 0.070$, thorax width: $P = 0.649$ and $P = 0.695$; ovipositor length: $P = 0.853$ and $P = 0.159$) among figs and trees, respectively. All components of these same data types were combined into one for subsequent analysis. Further, female *C. emarginatus* reared from *F. hainanensis* were significantly smaller than those reared from their typical host *F. auriculata*

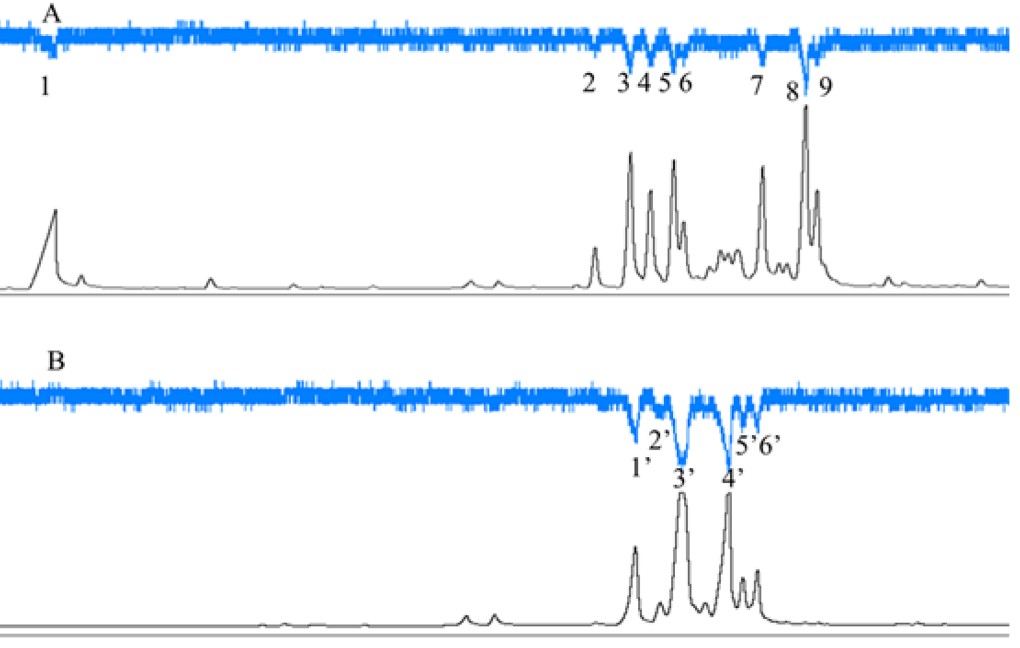

**Figure 5** Electroantennographic responses of *Ceratosolen emarginatus* to receptive fig scent extracts of (A) *Ficus auriculata* and (B) *F. hainanensis.* GC-FID (black line), and GC-EAD responses of C. emarginatus antennae (inverted blue line). VOC identification: 1: 2-Heptanone; 2: Ylangene; 3 and 1′: $\alpha$-Copaene; 4: $\alpha$-Funebrene; 5: $\alpha$-Gurgujene; 6 and 3′: $\beta$-Funebrene; 7: trans- $\beta$-Farnesene; 8: $\alpha$-Patchoulene; 9: $\beta$-Cadinene; 2′: $\alpha$-Cedrene; 4′: $\beta$-Cedrene; 5′: $\alpha$-Guaiaene; 6′: Aromadendrene).

**Table 2 Comparisons on the style length among figs and trees for *F. auriculata* and *F. hainanensis*.**

| *Ficus* species | Figs | Trees | Among figs | | Among trees | |
|---|---|---|---|---|---|---|
| | | | $\chi^2$ | $p$ | $\chi^2$ | $p$ |
| *F. auriculata* | | | | | | |
| Male figs | 32 | 4 | 635.99 | <0.001 | 584.53 | <0.001 |
| Female figs | 30 | 3 | 597.48 | <0.001 | 531.46 | <0.001 |
| *F. hainanensis* | | | | | | |
| Male figs | 30 | 3 | 68.81 | <0.001 | 16.09 | <0.001 |
| Female figs | 31 | 3 | 597.18 | <0.001 | 549.64 | <0.001 |

(head width: $P = 0.018$; thorax width: $P = 0.0002$), but similar to those *Ceratosolen* sp. associated with *F. hainanensis* (head width: $P = 0.925$; thorax width: $P = 0.999$). their ovipositor was shorter than those of both *C. emarginatus* and of *Ceratosolen* sp. ($P = 0.034$) raised on their respective typical hosts ($P < 0.001$; Table 4).

## DISCUSSION

Imperfect host-symbiont interactions, particularly in the case of highly species-specific pollination mutualisms, maintaining low rates of the wrong symbiont associating with the alternative host likely involve a fitness cost for both partners (*e.g.*, *Janzen, 1979*; *Ghana,*

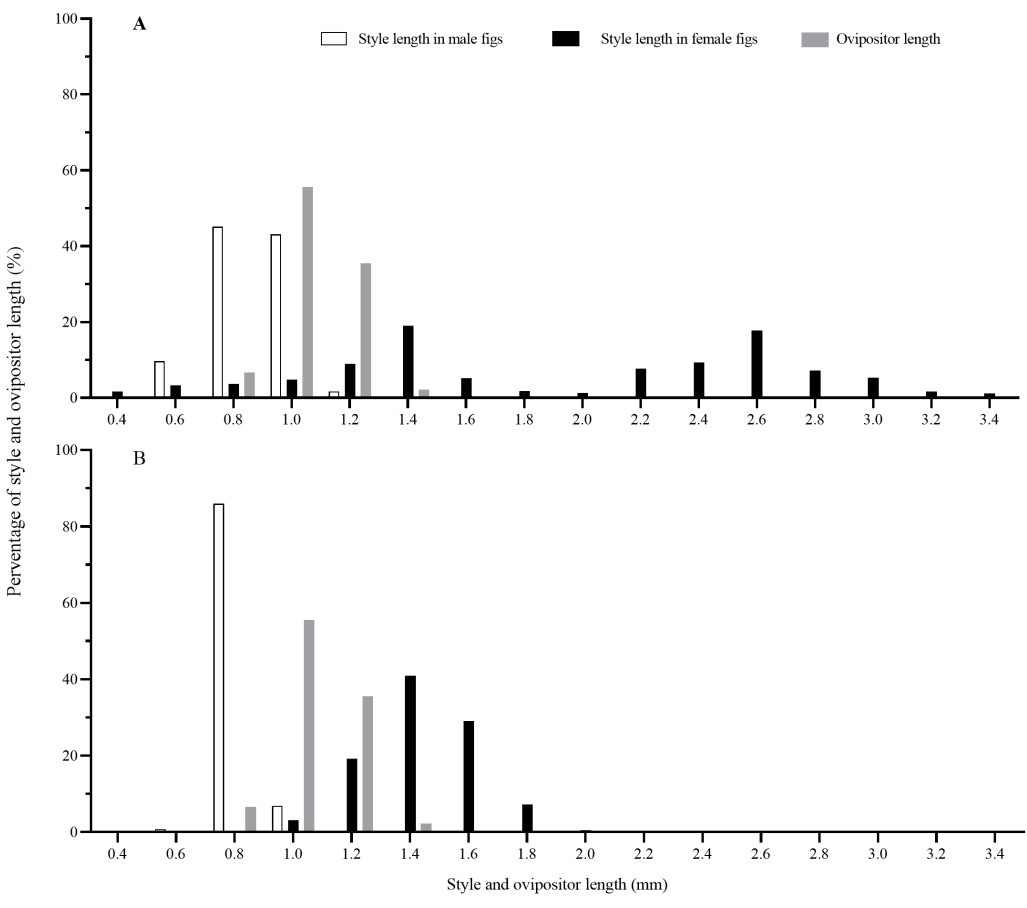

**Figure 6** **The distribution of fig style length and ovipositor length of *Ceratosolen emarginatus*.** (A) Style length of receptive figs from *Ficus auriculata*; (B) Style length of receptive figs from *F. hainanensis*.

**Table 3** **Consequences of offspring and seeds produced by single foundress of *Ceratosolen emarginatus* in *Ficus auriculata* and *F. hainanensis*.**

| Treatments | Sample | Abortion rates (%) | No. of offspring | | | No. of seeds | | |
|---|---|---|---|---|---|---|---|---|
| | | | Mean | SD | CV | Mean | SD | CV |
| Ce–Fa, male | 60 | 7.69 | 472.85 | 144.67 | 0.306 | | | |
| Ce–Fh, male | 60 | 41.67 | 618.49 | 211.55 | 0.342 | | | |
| Ce–Fa, female | 60 | 11.76 | | | | 974.21 | 642.48 | 0.659 |
| Ce–Fh, female | 60 | 100 | | | | | | |

**Notes.**
Abortion rate: the aborted figs to all sampling figs.
Ce, *Ceratosolen emarginatus*; Fa, *Ficus auriculata*; Fh, *F. hainanensis*.

*Suleman & Compton, 2015*). Moreover, given that these phenomena naturally occur, it is unknown what mechanisms impede either the formation of hybrid swarms among closely-related species or the evolution of hybridization-induced speciation events—both of which may cause breakdown of species-specificity with resultant fluxes in extant biodiversity patterns. Here we present a novel study that incorporates comprehensive sampling

**Table 4** Body size and ovipositor length of *Ceratosolen emarginatus* emerging from figs of the typical host *F. auriculata*, and of *Ceratosolen* sp. and *C. emarginatus* emerging from figs of alternative host *F. hainanensis*.

| Species | Sample | Head width (mm) (mean ± SD) | Thorax width (mm) (mean ± SD) | Ovipositor length (mm) (mean ± SD) |
|---|---|---|---|---|
| Ce | 77 | 0.48 ± 0.05 | 0.59 ± 0.05 | 1.15 ± 0.11 |
| Csp | 69 | 0.46 ± 0.04 | 0.55 ± 0.04 | 1.07 ± 0.11 |
| Ce from Fh figs | 114 | 0.46 ± 0.04 | 0.55 ± 0.07 | 1.03 ± 0.06 |

**Notes.**

Ce, *Ceratosolen emarginatus*; Csp, *Ceratosolen* sp.; Fa, *Ficus auriculata*; Fh, *F. hainanensis*.

alongside detailed ecological data to investigate both the mechanisms and potential evolutionary outcomes of alternative host-use events. We show that while similarity in semiochemicals of two *Ficus* species may facilitate alternative host-use, partner fidelity mechanisms may regularly operate that select against sustained introgression and help maintain species-specificity patterns.

## Pollinator visits to alternative hosts

Frequent use of several host species by a pollinator species has been reported in a few cases for American and African monoecious *Ficus* lineages (*Cornille et al., 2012*; *Machado et al., 2005*; *McLeish & Van Noort, 2012*) and in some Asian dioecious *Ficus* lineages (*Su et al., 2022*; *Wang, Cannon & Chen, 2016*; *Yu et al., 2022*). In some of these cases olfactive messaging has converged showing that pollinator sharing is a plant adaptation (*Cornille et al., 2012*; *Wang, Cannon & Chen, 2016*). In our study, we investigated a case in which a pollinator species, *C. emarginatus*, has a main host and alternative host. The two host species' populations, were located several kilometers apart in their natural habitat. We identified to species 3,021 wasps that had entered 293 receptive phase figs and 1,400 offspring wasps emerging from 140 figs. Our results reveal strong specialization with a low frequency of individuals colonizing the alternative host species, as seen in some other dioecious *Ficus* species (*Moe, Rossi & Weiblen, 2011*; *Silvieus, 2006*; *Weiblen, Yu & West, 2001*). Further, the typical host *F. auriculata* was predominantly visited by its own associated pollinators, *C. emarginatus*.

We document a reduction in the proportion of alternative pollinator offspring relative to the initial proportion of foundresses in natural populations. Moreover, female *C. emarginatus* experimentally introduced into figs of *F. hainanensis* produced larger broods than on its typical host and more than is produced by *F. hainanensis*'s typical pollinator, *Ceratosolen* sp. (*Yang et al., 2012*). However, our results suggest that *C. emarginatus* individuals visiting figs of *F. hainanensis* have reduced fitness due to trait mismatch resulting from reduced offspring ovipositor length (Table 2) that is likely to compromise oviposition ability (*Liu, Yang & Peng, 2011*; *Yang et al., 2012*) and due to the reduced fecundity associated with reduced size (*Moore & Greeff, 2003*). It is possible that individuals of *C. emarginatus* visiting figs of *F. hainanensis* are wasps that have failed to locate receptive figs of *F. auriculata* (*Liu et al., 2015*), and have become less choosy towards the end of their life spans. In some cases, wasps entering alternative hosts failed to reproduce. Additionally, our results show that 41.67% of receptive male *F. hainanensis* figs harbored pollinating

wasp *C. emarginatus*. This is reminiscent of the situation for the pollinators of *F. hirta* and *F. triloba* in Guangdong province, China, though in that case fitness consequences were not investigated (*Yu et al., 2022*). Co-occurrence of the pollinator species across two fig species indicates potential for hybridization, although we did not observe any wasps possessing morphological characters suggestive of hybridization. In *F. rubiginosa*, three cryptic species of *Pleistodontes imperialis* coexist in the same localities while retaining reproductive isolation (*Sutton et al., 2017*), with *Wolbachia* identified as the most likely candidate of post-zygotic reproductive isolation (*Haine & Cook, 2005*). Similarly, no hybridization was detected between pollinating-wasps of genus *Pegoscapus* in Panama (*Satler et al., 2022*).

## Host volatile semiochemicals attracting pollinators

It has been suggested that closely related sympatric figs species may emit similar floral scents to attract pollinators and this could result in pollinators confusing typical and alternative host species (*Moe, Rossi & Weiblen, 2011*; *Wang, Compton & Chen, 2013*). The VOC composition of the receptive fig odours of *F. auriculata* and *F. hainanensis* were clearly differentiated, consistent with the different VOC profiles between sister species of section *Papuacyse* and between subspecies of *F. trichocerasa* in Papua New Guinea (*Souto-Vilarós et al., 2018*), and between *F. hirta* and *F. triloba* (*Yu et al., 2022*). They shared 34 VOCs including 10 quantitatively important compounds (*i.e.,* >5%), but only two that were shown to elicit wasp antennal response as semiochemicals. The shared semiochemicals eliciting antennal response represented 19% of receptive male fig odours and 13% of receptive female fig odours in *F. auriculata* and 38% and 31% for male and female *F. hainanensis* figs respectively, indicating a large degree of chemical attractant overlap. Wasp attraction to receptive figs may be from long and short distance alongside contact attractants. In *F. curtipes* it has been shown that one VOC is mainly responsible for long distance attraction while another is more important for fig entry (*Gu & Yang, 2013*). When *C. emarginatus* were given a choice between *F. auriculata* and *F. hainanensis* receptive figs, 30% chose *F. hainanensis* figs, a figure much higher than the observed frequency of alternative pollinators on our wild figs and suggesting either that initial contact with typical host odour in the stem of the Y-tube may result in reduction in choosiness.

## Offspring of pollinating fig wasps in alternative host species

*C. emarginatus* produced more but smaller offspring in *F. hainanensis* than in its typical host (Table 3). *F. auriculata* and *F. hainanensis* fit the general pattern by which style lengths in male figs are shorter than the ovipositor of pollinators (Fig. 6), allowing wasps to oviposit into pistillate flowers of male figs (*Ganeshaiah et al., 1995*; *Shi, Yang & Peng, 2006*; *Weiblen, 2002*). *C. emarginatus* ovipositors are longer than the styles in male figs of *F. hainanensis* which have shorter styles than *F. auriculata*. This may explain why *C. emarginatus* produced more offspring in *F. hainanensis* than in *F. auriculata*, as oviposition can be more rapid in flowers with shorter styles. However, offspring body size was reduced in comparison with *C. emarginatus* developing in *F. auriculata* (Table 4) and body size is known to correlate with fecundity in fig pollinating wasps (*Moore & Greeff, 2003*). This probably results from the developing larva consuming the endosperm of the smaller plant ovules of *F. hainanensis*

(0.61 ± 0.01 mm), compared with those of *F. auriculata* (0.88 ± 0.01 mm), which may cause a concomitant reduction in fitness due to reduced body size featuring shorter ovipositors which should impinge subsequent oviposition attempts (*Moore, Pienaar & Greeff, 2004*; *Stone et al., 2002*). Nevertheless, *C. emarginatus* offspring will leave male *F. hainanensis* figs carrying pollen. These wasps are likely to search for *F. auriculata* figs and may subsequently initiate host hybridization (*Machado et al., 2005*). However, small wasps will probably have reduced capacity to reach receptive figs and will therefore be poor pollen vectors. In comparison, smaller sized *C. emarginatus* developing in the alternative host *F. hainanensis* may be fully capable of entering and ovipositing in *F. hainanensis* figs. If their parents had entered the atypical host because of a mutation in its olfactory system, or if host choice ability is formed through host environmental effect, then the evolution of a new host preference is possible. Such variant *C. emarginatus* are likely to be outcompeted by *Ceratosolen* sp. Only in locations where *Ceratosolen* sp. is absent would these genetic variants thrive and progressively adapt to the new host. This may happen, as populations of *F. hainanensis* are generally small and localized. The case of *C. emarginatus* also visiting figs of *F. oligodon* is somewhat different as the two host species produce identical receptive fig odours suggesting selection on the two host species to mimic each other and attract a same population of pollinating wasps (*Wang, Cannon & Chen, 2016*).

In experimental introductions into receptive figs of *F. hispidioides* of wasps originating from four other *Ficus* species, no wasps developed, although oviposition attempts led to ovule development in male figs (*Moe, Rossi & Weiblen, 2011*). Beyond the *F. auriculata* species complex which includes *F. auriculata*, *F. oligodon* and *F. hainanensis*, successful emergence of offspring wasps of a same species from two or several locally co-occurring alternative host figs has been reported for *Elisabethiella stuckenbergi* visiting *F. burkei* and *F. natalensis*, subgenus *Urostigma* in South Africa (*Cornille et al., 2012*) and for *Blastophaga silvestriana* visiting a number of species of subgenus *Ficus*, section *Ficus* in China (*Su et al., 2022*). In this later case some genetic data suggests the presence of hots races (*Wachi et al., 2016*), while other data suggests that a same population of wasp visits several host species (*Su et al., 2022*). Nevertheless, none of these studies includes the experimental confirmation we provide here on successful offspring development in alternative host. Hence, despite reports of fig pollinating wasps visiting several host species the outcome of these visits should be ascertained before drawing conclusions on their evolutionary significance (*Moe, Rossi & Weiblen, 2011*). Mechanisms can therefore exist that allow alternative interactions while simultaneously preventing introgression and species-specificity break-down. However, it can be further envisaged that this system could be easily reconfigured to favour hybridization events if partner fidelity mechanisms fail or are selectively removed, potentially due to changes in abiotic conditions and local extinctions of the population of pollinators of a potential alternative host.

## Hybridization implied by pollinator's presence in an alternative host

With respect to potential plant hybridization, our data show that all female *F. hainanensis* figs visited by *C. emarginatus* undergo selective abortion (Table 3). High abortion levels (88.3% and 100% in male and female figs, respectively) have also been documented

for *F. auriculata* figs visited by wasps (most probably *Ceratosolen* sp.) emerging from *F. hainanensis* (*Yang et al., 2012*). As viable hybrids have been observed between phylogenetically more distant *Ficus* species (*e.g.*, *Condit, 1950*), we suggest that the lack of seed production results from variation between wasp species in pollination behavior and trait matching. Indeed, artificial pollination leads to the production of viable offspring (*Wei et al., 2014*).

Four genetic studies based on RFLPs (*Parrish et al., 2003*) and on microsatellite data (*Moe, Rossi & Weiblen, 2011*; *Wang, Cannon & Chen, 2016*; *Wei et al., 2014*) have suggested the presence of hybrids between closely related *Ficus* species. In two of those studies, genetic results confirmed that morphologically intermediate individuals were interspecific hybrids (*Parrish et al., 2003*; *Wei et al., 2014*). We cut open all aborted female figs to inspect the ovules in every cavity containing a *C. emarginatus* foundresses. We observed no swelling of the ovules. This indicates that strong pre-zygotic isolation mechanisms prevent the hybridization of *F. auriculata* and *F. hainanensis*.

Interestingly, although our experiments suggest that *F. auriculata* and *F. hainanensis* do not typically hybridize, some hybrids have been documented between them (*Wang, Cannon & Chen, 2016*). *F. oligodon* which is sympatric with both species hybridized with *F. auriculata* and produced a normal number of seeds (*Yang et al., 2012*), but hybridization events with *F. hainanensis* produced few seeds and a high fig abortion ratio (Table 3). This suggests that *F. oligodon* may act as a bridge species facilitating introgression between *F. auriculata* and *F. hainanensis*. Therefore, in addition to sharing pollinators, such a bridge species may play a pivotal role in speciation events and co-diversification with some wasp exchanges between closely related *Ficus* species (*Cornille et al., 2012*).

## CONCLUSIONS

Our findings show that a low frequency of *C. emarginatus* associated with the typical host *F. auriculata* enter the alternative host *F. hainanensis* in natural populations, and that a small proportion of wasps' offspring complete their development in the alternative host. *C. emarginatus* appear to enter receptive figs of the alternative host because emitted signal scents share some of the main semiochemicals, although their overall VOC compositions are different. However, mismatches between the length of ovipositors of fig wasps and of styles in male figs appear to limit successful reproduction of pollinators in the alternative host, *F. hainanensis*, and further reduces the body size of progeny that are produced. Our findings also show that no seeds were produced when *C. emarginatus* with *F. auriculata* pollen were introduced into female figs of *F. hainanensis*, owing either to mismatches between wasp behavior and plant anatomy, or to interspecific pre or post zygotic incompatibility, or both. Thus, we show that despite a potential for multiple host use and genetic introgression, specificity in sympatric fig-wasp pollination mutualisms can be maintained. Moreover, our data indicate that suites of mechanisms either promoting or hindering hybridization may co-exist, with eventual outcomes contingent on dominant partner fidelity mechanisms. However, over evolutionary timescales, occasional use of alternative hosts may provide opportunities for host shifts, hybridization or other events leading to diversification of both figs and wasps if partner fidelity mechanisms fail or are selectively removed.

## ACKNOWLEDGEMENTS

We appreciate the team of the Yanqiong Peng's Laboratory of Xishuangbanna Tropical Botanical Garden (XTBG, CAS) for enthusiastic and superb technical support. We also thank Doyle Benton McKey to for very useful comments on the manuscript. We also thank Ming-Xin Liu, Xiao-Mei Liu and Ling-Ru Wang for help in carrying out field experiments and counting offspring.

### Funding

This research was supported by grants from the National Natural Science Foundation of China (Nos. 31760107 and 32160296), and the Young Top-Notch Talent of High-Level Cultivation in Yunnan Province (YNWR-QNBJ-2018-131 and YNWR-QNBJ-2019-123). The funders had no role in study design, data collection and analysis, decision to publish, or preparation of the manuscript.

### Grant Disclosures

The following grant information was disclosed by the authors:
The National Natural Science Foundation of China: 31760107, 32160296.
The Young Top-Notch Talent of High-Level Cultivation in Yunnan Province: YNWR-QNBJ-2018-131, YNWR-QNBJ-2019-123.

### Competing Interests

The authors declare there are no competing interests.

### Author Contributions

- Hua Xie performed the experiments, analyzed the data, prepared figures and/or tables, authored or reviewed drafts of the article, and approved the final draft.
- Pei Yang conceived and designed the experiments, performed the experiments, analyzed the data, prepared figures and/or tables, authored or reviewed drafts of the article, and approved the final draft.
- Yan Xia performed the experiments, analyzed the data, authored or reviewed drafts of the article, and approved the final draft.
- Finn Kjellberg analyzed the data, authored or reviewed drafts of the article, and approved the final draft.
- Clive T. Darwell analyzed the data, authored or reviewed drafts of the article, and approved the final draft.
- Zong-Bo Li conceived and designed the experiments, analyzed the data, prepared figures and/or tables, authored or reviewed drafts of the article, and approved the final draft.

### Field Study Permissions

The following information was supplied relating to field study approvals (i.e., approving body and any reference numbers):

# PeerJ

Agonid wasps, *Ceratosolen emarginatus* and *Ceratosolen* sp., are common insects and collection was permitted by the leader of Department of Horticulture and Landscape, Xishuangbanna Tropical Botanical Garden, Chinese Academy of Science.

## Data Availability

Raw data are available in the Supplemental Files.

## Supplemental Information

Supplemental information for this article can be found online at http://dx.doi.org/10.7717/peerj.13897#supplemental-information.

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
