# Peer review of "Maintenance of specificity in sympatric host-specific fig/wasp pollination mutualisms"

_PeerJ, doi:10.7717/peerj.13897_

## Round 0.1 · original submission · Major Revisions

Dear Dr. Xie and colleagues:

Thanks for submitting your manuscript to PeerJ. I have now received three independent reviews of your work, and as you will see, the reviewers raised some concerns about the research. Despite this, these reviewers are quite optimistic about your work and the potential impact it will have on research studying fig-wasp mutualism. Thus, I encourage you to revise your manuscript, accordingly, taking into account all of the concerns raised by all three reviewers.

While the concerns of the reviewers are relatively minor, this is a major revision to ensure that the original reviewers have a chance to evaluate your responses to their concerns. There are many suggestions, which I am sure will greatly improve your manuscript once addressed.

There are many minor suggestions to improve the manuscript (typos, nuances, etc.). Reviewer 1 has kindly provided edits on your manuscript.

Therefore, I am recommending that you revise your manuscript, accordingly, taking into account all of the issues raised by the reviewers.

I look forward to seeing your revision, and thanks again for submitting your work to PeerJ.

Good luck with your revision,

-joe

·

Basic reporting

1. Basic reporting: The manuscript is written quite clearly, in grammatical English; there are a couple of typographical errors.
Line 101: Correct to asympatrically
Line 357: Correct to where
All the elements of a professionally written paper are evident including the raw data. Figures and tables are also appropriate

Experimental design

2. Experimental Design: This study was designed to examine the possibility of hybridisation between two relatively closely related and sympatric fig species. If hybridisation does not occur, then what are the likely mechanisms of reproductive isolation between the species? The authors have conducted a host of appropriate experiments and observations to answer this question. The experimental design is therefore adequate. The methods are provided in adequate detail to allow for replication. The article is well within the journal's scope

Validity of the findings

3. Validity of the findings: The findings are well documented and the tests of the hypotheses have been adequately conducted. It is clear that there are reasons for reproductive isolation which the authors have clearly demonstrated.

Additional comments

In general, this is a well conducted study. I have a few suggestions about the ordering of some text which I have annotated in the accompanying pdf.

Reviewer 2 ·

Basic reporting

No comment

Experimental design

No comment

Validity of the findings

No comment

Additional comments

This paper reports laboratory and field experiments on host-symbiont specificity in two Chinese fig-wasp mutualisms in which the host trees are members of a three-species complex. The data set is impressive and the authors are to be congratulated on the presentation of a comprehensive set of results ranging from the analysis of plant volatiles to behaviour of the pollinating wasps.

Overall, the work is fairly well-written and presented but suffers throughout via overly long text and frequent use of unnecessary jargon. I think the paper would be much improved by a sentence-by-sentence revision by a fluent English speaker. Additionally, the verbose use of language reduces clarity, especially in the introduction, meaning that it is not entirely clear why specific data were collected and why other data were not recorded (see below). Finally, it needs to be acknowledged that the paper will be read by a general reader and some text needs more detailed explanation. At the moment, I suspect that a non-specialist may not fully understand some text, why certain experiments and comparisons were made, and the interpretation of some of the results.

I think the data presented will certainly form the basis of a very interesting paper and fits the remit well for a general journal such as PeerJ. However, I think an extensive revision is required.

Specific comments

Why is the text not double spaced? Single spaced text makes things more difficult for reviewers, especially those who like to annotate a hard copy with red ink.

Abstract

The final sentence of the abstract is a good example of the use of excessive verbiage and jargon. Please simplify.

Line 39. What is a ‘diversity flux’?

Introduction

41-42 The first sentence is unclear. What unique components? Pollinator shifts between hosts seem to me to be more likely to promote hybridisation and gene flow rather than speciation.

51 Private code for what? This is vague.

51-59 Do you mean in different nursery pollination mutualisms there is evidence of host-symbiont coevolution and different host species produce their own VOCs especially attractive to their own pollinator species?

60 If there is strict one-to-one specificity, what causes the split? Allopatric speciation?

65 Is there any evidence of hybrid pollinating fig wasps?

86 There needs to be a less ambiguous description of the differences between monoecious and dioecious Ficus.

87 Receptive to what?

91 This is so only if 100% effective but nothing is perfect in nature, there is always variation.

92-95 A general reader is unlikely to fully understand this. More background detail is required.

103 Speculation or hypothesis?

121 If pollinated by ‘incorrect’ pollen, a fig may be more likely to abort, especially a female fig in a dioecious species. A male fig tree may still benefit by producing wasps that will disperse (hybrid) pollen. See Zhang et al. 2019. Ecology.

126 iii) Why was the reciprocal experiment not conducted using the opposite wasp/host combination? This needs to be explained as the question comes to mind several times when reading the paper. Also, why were seeds/galls not counted, nor fig abortion rates measured? High rates of abortion can exert strong selection on wasps to pollinate ‘correctly.’

M & M

134 Typo in latin name

134-136 Are the figs of the two species of similar size? I am aware that F. aur has very large figs, and there is a strong positive relationship between wasp and fig size among Ficus. If there is a size difference between the two systems, then due to correlated traits there will be a predictable difference in e.g. wasp head width, thorax width length, ovipositor length, all of which may affect the likelihood of a wasp entering a fig through the ostiole and thus the potential for hybridisation. All of this needs to be explained.

143 Debate? References needed. Do you mean ‘unclear’?

147 Traditionally? Or ‘usually’?

161 How were wasps identified? Characters?

162 How many figs for each tree per site? For data analysis, could a GLMM be used for e.g. wasp traits using site/tree as random factors to eliminate sampling error?

164 For figs of different stages were the same or different trees used?

170 Fig character differences? Pollen, flowers?

179 How were wasps collected/stored?

186 Detail on wasp and fig sizes required.

193 Between what?

196 How many wasps made no decision?

201 Sex not gender.

201 N Figs?

226 Why was only one ‘incorrect’ wasp / host Ficus combination tested and not the reciprocal combination? This needs to be explained.

240 Unclear to a non-specialist why this was done. More information on the natural history and general biology in the methods would help here.

246 How was randomisation achieved? Or was selection ‘haphazard’ rather than truly random.

249 More detail required on the measurements taken. Magnification? Was the ovipositor sheath removed? If wasp size measured it would need to be explained why, see above. I do not understand why fig traits were also not measured other than style length.

254 What is meant by ‘young figs’? What was the source of the wasps, and hence the source of the pollen? Seed counts, abortion rates (abortion needs to be clearly defined as this means different things to different people, especially botanists).

273 No mention of head width, thorax width before now so this just comes out of no-where. An explanation of fig size, wasp size and how these vary and may affect the direction of gene flow between trees and possibly wasps needs to be included.

Results

282 Contain offspring or foundresses? Please clarify.

316 Which VOCs were chosen to be investigated in more detail and why? This section is a little vague. Why were Ceratosolen sp. not tested?

327 What does ‘somewhat’ mean here? More clarity is needed regarding statistical significance (or not). Wasp size, fig size differences?

333. Own ‘proper’ host or ‘other, incorrect’ host? Please clarify.

335. Fig abortion seems really important here but no mention in the introduction or the methods. In the context of fig-wasp systems it needs to be defined and if possible, data presented as this could clearly act as a strong agent of selection in maintaining ‘correct’ pollination/oviposition behaviour in symbiont wasps.

337 This is interesting. More clarity is needed regarding differences own v incorrect host. Also interesting regarding size but more detail is needed. Ostiole sizes regarding a filter to wasps of the ‘correct’ size?

The results need to be clearer and more succinct throughout. I think if a small natural history section is added to the methods things would be a lot clearer especially to a non-specialist. If the introduction were made clearer and some specific predictions made, then I think it would be easier to produce a more succinct and clearer results section.

Discussion.


342 Imperfect host-symbiont alignment is probably very common and can be expected in an imperfect natural world. I think it would help to emphasise this point in the introduction and then go on to pose the question as to why we see such low rates of the ‘wrong’ symbiont associating with any particular host.

351 Promoting v inhibition – vetoing does not appear to occur as there are always a few ‘mistakes.’

342-353 Please simplify the language and eliminate jargon wherever possible.

359 Natural rates of visitation were not recorded. Foundresses were counted but this is not the same as visitation because the wasps have visited a fig and then entered it. It would be most interesting to present visitation data though of correct v wrong wasp species on each of the single host fig species and how this does or does not differ to the patterns of foundresses within figs.

364. Still, not 100% precision but that is to be expected. We expect ‘mistakes’ and hence variation in such systems.

365 I am still puzzled as to why experiments using Ceratosolen sp. were not conducted.

373 Evidence for variation in body size per se and oviposition in agaonid fig wasps?

380 It would be interesting to know if the patterns of egg deposition, i.e how many eggs on average were laid by each foundress, varied according to wasp species and the species of fig it was ovipositing into.

381 In some Ficus with cryptic pollinator species, e.g. F. rubiginosa, different wasp species can reproduce in the same figs. If this happened in the species complex under scrutiny here, then the ‘wrong’ pollinator species may be able to successfully reproduce in the incorrect host if the host fig was pollinated and galled by another foundress of the correct species. E.g. the fig may be less likely to abort and the incorrect species wasp would ‘free-ride’ on the ‘correct’ species foundress. Of the figs collected, how many containing wasps of the ‘wrong’ species also contained a foundress of the ‘correct’ species?

389 Which sister species? Who are ‘they’?

419 References for this? Are there differences in abortion rates between males and females of the same species and do these vary according to which species of wasp entered and pollinated/oviposited? (see Zhang et al. 2019).

436 Seed data for all wasp-tree combinations?

438 Male figs should have lower abortion rates than female figs when they contain only the ‘wrong’ pollen.

448 Why not unaborted figs? What were the abortion rates?

454 This would have been best mentioned in the introduction.

465 Progeny fitness was not measured so this statement goes beyond what the data show.

472 Mechanisms presented show likely inhibition of gene flow between host species and thus against hybridisation. Which mechanisms presented will facilitate hybridisation? I do not think any convincingly show this here.

Figure 2 Is this figure necessary? If removed it would not detract from the story being told.

Table 1 Seed count data are interesting here but there is no mention anywhere in the paper of this. Also, reciprocal data are needed as are data for both host-wasp ‘correct’ combinations to act as natural controls.

Reviewer 3 ·

Basic reporting

Fine

Experimental design

Fine

Validity of the findings

Fine

Additional comments

I enjoyed reading this interesting and thought-provoking paper. The topic of host specificity in mutualisms is of considerable ecological and evolutionary importance but there have been few studies of the consequences of atypical host use in specialist nursery pollinator mutualisms. I commend the authors on the use of a range of different methods in investigating this issue, including field surveys, characterisation of volatile attractants, behavioural responses and reproductive outcomes. Overall, this is a well-written and structured paper, reporting a well-executed study. My comments are mostly specific points requiring some clarification of details, or further consideration of arguments / explanations.

Terminology. I suggest the use of alternate host (as used in the conclusion) over non-typical host (as used elsewhere). I think this would be more transparent to the reader e.g. on line 19 in the Abstract. I am also not keen on the use of the word vetoing as in line 45. I think that alternate wording such as partner choice mechanisms or partner fidelity mechanisms would be clearer.

Line 46. Why did they have the potential to undermine extend biodiversity patterns? Surely the events themselves are part of the current biodiversity patterns that you document and underlying mechanisms help explain the existing patterns?

60. Strict specificity… leads to one-to-one interaction patterns. This is a circular argument.

82. Looks like a typo here. Should the sentence start “Ficus species…” ?

91. “thus ensuring..”. But they don’t, as you show in this study. “Promoting” is a better word here.

143. As the taxonomic status is still under the debate, can you confidently identify trees as belonging to one species or the other and how?

153. If I understand correctly, the wasps can only be identified to species under the microscope. If so, how did you get female wasps of known species to introduce to figs?

162. Did you collect ripe and immature figs at the same time such that they represent different cohorts, or were the ripe figs collected later and represent the same cohort of figs? This distinction influences how you interpret the comparison of the situation in immature and mature figs?

278. It would be interesting to also test if the distribution of pollinators entering the alternate host fig is random across figs / trees.

333. It’s ery interesting to see that wasps produced more offspring (even if smaller) in the alternate host spcies. This could have significant evolutionary consequences and shows that that wasp size is strongly influenced by host environmental effect. The host shifting wasp becomes much more similar in body size to the normal pollinator of the alternate host. The sample size for these measurements is shown in Table 2 and is large, but it would also be interesting to know how variable it is across figs of the same species (and if correlated with fig size if you have data).

367-8 See query about fig cohorts above (line 162).

373 Is there here (or elsewhere) any evidence that smaller individual pollinator wasps have “compromised oviposition abilities”?

414 related to line 373 point. Be clearer about which host fig you are discussing here. Smaller size of C. emarginatus developing in wrong host (Fh) may decrease their success if entering the ancestral host (Fa) but not if they now seek out Fh, as these wasps are about the same size as the usual Fh pollinator.

442. Seem to be words missing here…

Figure 7. It would be informative to show the distribution of ovipositor lengths for both wasp species in each chart alongside the host fig style length distributions.

---

## Round 0.2 · accepted · Accept

Dear Dr. Xie and colleagues:

Thanks for revising your manuscript based on the concerns raised by the reviewers. I now believe that your manuscript is suitable for publication. Congratulations! I look forward to seeing this work in print, and I anticipate it being an important resource for groups studying fig-wasp mutualism. Thanks again for choosing PeerJ to publish such important work.

Best,

-joe